# Early temperament and physical health in school-age children: Applying a short temperament measure in a population-based cohort

**Jennifer Chun-Li Wu[1], Pei-Ling Wang**[2]*, **Tung-liang Chiang**[3]

**1** Department of Early Childhood and Family Education, National Taipei University of Education, Taipei, Taiwan, **2** Department of Early Childhood Education, University of Taipei, Taipei, Taiwan, **3** Institute of Health Policy and Management, National Taiwan University, Taipei, Taiwan

* peilinwang2016@gmail.com

## Abstract

Temperament has drawn considerable attention in the understanding of behavioural problems and psychopathology across developmental stages. However, less of a focus has been placed on the role of temperament in physical aspects of health. We aimed to examine the relations between early temperament traits and physical health in school-age children. This study used longitudinal data of 18,994 children (52.4% boys) born in 2005 from the Taiwan Birth Cohort Study, in which follow-up surveys were conducted via face-to-face interviews with the child's caregiver. Temperament at 5.5 years of age was assessed using a nine-item measure, and two higher-order temperament traits, surgency and regulation, were derived through confirmatory factor analysis. Physical health outcomes at age 8 included caregiver-rated general health status and medically attended injuries. Multiple logistic regression analysis was applied, with the child's birth outcome, early health status or injury history, health behaviours and family socioeconomic status as control variables. The results indicated that higher levels of surgency and regulation, as early temperament traits, significantly predicted lower odds of caregiver-rated poor health in later years. Higher level of regulation was also associated with lower odds of injury risk. Our findings suggest that assessing early temperament traits could be useful for the promotion and management of physical health in young school-age children.

## Introduction

Temperament is generally defined in terms of early-appearing individual differences that manifest in the aspects of activity, emotionality, sociability and regulation, and tend to be consistent in individuals across situations. While temperament is innate, with a heredity origin and biological roots, changes in temperament do occur due to maturation and environmental influences [1, 2]. Previous research has applied a variety of measurements and approaches to generate constructs of child temperament based on different theoretical models. These include

**Data Availability Statement:** The Taiwan Birth Cohort Study is fully funded by the Health Promotion Administration, Ministry of Health and Welfare (MOHW). Data are owned by the MOHW

and contain personal and health information. Data access requests should be directed to the Health and Welfare Data Science Center, Department of Statistics, MOHW (https://dep.mohw.gov.tw/dos/np-2497-113.html) in charge of data management and application review.

**Funding:** The birth cohort study was funded by the Health Promotion Administration, Ministry of Health and Welfare (DOH97-HP-1702) and TL Chiang is the Principal Investigator. The funder had the role in study and data collection.

**Competing interests:** The authors have declared that no competing interests exist.

Thomas and Chess's behavioural style model and the core assumption of underlying psychobiological processes in individual reactivity and regulation by Rothbart and colleagues, to name a few [3, 4].

For the past few decades, a large body of research has documented the roles of early temperament in shaping developmental outcomes such as socioemotional competence [5, 6], cognitive ability [7], pre-academic skills and school readiness [8, 9]. In their landmark investigation of temperament, Thomas and Chess observed that temperament traits, although normal in and of themselves, can give rise to children's behavioural problems, especially if children's temperament traits are poorly matched with the care environment [10]. Since then, there has been an exponential increase in the empirical literature that has advanced our understanding of temperament traits in relation to developmental psychopathology [11]. Regarding outcomes in preschool and school-age children, certain early temperament traits (e.g., low effort control and high activity level) have been found to be associated with externalising behaviours and the risk for attention-deficit hyperactivity disorder, while other traits, such as high irritability and shyness, have been reported to predict internalising problems, anxiety or depression [3, 4].

With the extant literature on temperament predominantly in the fields of human development and psychology, comparatively less research has taken a temperament perspective in examining children's physical health. The primary health outcomes of interest in these studies were disease-related health conditions and injury. For instance, higher activity level [12], lower attention span in boys and higher soothability in girls [13] and difficult temperament [14] were found to be associated with greater obesity risk. Ravaja and Keltikangas-Järvinen's follow-up study demonstrated that early positive emotionality predicted a lower level of metabolic syndrome precursors in children, while negative emotionality increased the risk [15]. In addition, younger school-age children who exhibited higher activity levels and less controlled behaviours [16] or poorer effortful control [17] were prone to unintentional injuries. However, very little is known about the connection between temperament and children's general health. A recent study showed that higher emotionality in toddlers, characterized by being easily and intensely aroused, was predictive of poorer mother-rated overall health status [18]. Linking temperament traits and physical health is of particular importance for children during the formative years of life. One major reason is that the investigation of temperament influences could broaden our knowledge about the early signs of disease onset or processes [19]. Considering various behavioural style traits in clinical work with children to identify signs of functioning problems and possible pathology has also been advocated [20]. Moreover, certain temperament-related behaviours represent major components of unintentional injury risk, which has been one of the primary child health issues in the developed world. Thus, our study aimed to examine the relationship between early temperament traits and the physical health (i.e. general health status and injury occurrence) of young school-age children, using longitudinal data from the Taiwan Birth Cohort Study (TBCS). Toward that end, we first validated a short temperament measure that was designed and incorporated as part of the TBCS survey in early childhood.

As per our review, most of the studies linking early temperament to later psychopathology or health outcomes have employed a dimensional approach to temperament, be it the use of lower-order or higher-order traits. Generating broader traits from particular lower-order dimensions that tend to vary together (e.g., activity level and approach) is consistent with the more recent notion that temperament is organized hierarchically [2, 4], and also increases the statistical power of temperament predicting other variables under study [3]. Taken together, we used higher-order temperament factors in this study to investigate the association with children's physical health.

## Methods

### Construction of the short temperament measure

The short temperament measure developed for use in the TBCS was based on the nine temperament dimensions identified in the New York Longitudinal Study (NYLS) by Thomas and Chess–*activity*, *rhythmicity*, *approach*, *adaptability*, *intensity*, *mood*, *persistence*, *distractibility*, and *threshold of responsiveness* [10]. One generic item was designed to reflect the core concept of each temperament dimension on a 7-point Likert scale. For example, the item "Your child is energetic and physically active, OR is quieter and does not enjoy games involving physical movement." was used for the activity dimension and to be rated from 1 (low activity level) and 7 (high activity level). For the distractibility dimension, we mainly captured the aspect of attentional focusing but not emotional distractibility by asking "Your child can concentrate on doing a task, OR is easily disturbed by external stimuli then distracted." We renamed this as the *attention* dimension. In addition, we asked a reverse-scored question about the level of *sensitivity* to measure the sensory threshold. All item statements were initially written by the second author with reference to the Behavioural Style Questionnaire (BSQ), a NYLS-based instrument measuring temperament in 3- to 7-year-old children [21]. The final version of items (S1 Table) were further reviewed and confirmed by the TBCS research team.

The validation of the nine-item brief measure was performed by correlating the single item scores with their corresponding subscale scores of the BSQ. The Taiwanese version of the BSQ has been validated through back-translation and pre-test procedures, with subscale test-retest reliability coefficients between 0.72 and 0.92 [22]. We conducted a pilot study in 2010 with the parents of a convenient sample of 228 preschool children (age range: 36 to 84 months; mean age = 59.8 months; 48.2% males) from two cities in northern Taiwan. The pilot questionnaire contained three parts in the order of demographic questions, the short temperament measure and the validated BSQ. Table 1 presents the item mean scores of the TBCS short measure, BSQ subscale mean scores as well as the correlations between the single items and the BSQ subscale scores. Except for the attention dimension, all correlation coefficients were statistically significant (p < .001) with the strength ranging from 0.36 to 0.66, providing evidence for the applicability of the nine-item temperament measure. A non-significant correlation in attention

**Table 1. Descriptive statistics and correlation coefficients between the TBCS items and the BSQ subscale scores (N = 228).**

| TBCS items | | | BSQ subscales | | | | | | | | | | | | | | | |
|---|---|---|---|---|---|---|---|---|---|---|---|---|---|---|---|---|---|---|
| | | | Activity | | Rhythmicity | | Approach | | Adaptability | | Mood | | Distractibility | | Persistence | | Sensory threshold | | Intensity | |
| | | | *M* | *SD* | *M* | *SD* | *M* | *SD* | *M* | *SD* | *M* | *SD* | *M* | *SD* | *M* | *SD* | *M* | *SD* | *M* | *SD* |
| | *M* | *SD* | 3.78 | 0.91 | 4.52 | 0.86 | 4.33 | 0.92 | 4.98 | 0.85 | 4.88 | 0.66 | 4.54 | 0.80 | 4.02 | 0.59 | 3.14 | 0.81 | 3.99 | 0.83 |
| Activity | 5.31 | 1.25 | **0.46**[***] | | -0.29[***] | | 0.26[***] | | 0.10 | | 0.14[*] | | -0.36[***] | | -0.34[***] | | -0.22[**] | | 0.24[**] | |
| Rhythmicity | 5.39 | 1.33 | -0.08 | | **0.51**[***] | | -0.03 | | -0.04 | | 0.16[*] | | 0.29[**] | | 0.15[*] | | 0.20[*] | | -0.11 | |
| Approach | 4.24 | 1.63 | 0.27[***] | | -0.11 | | **0.66**[***] | | 0.62[***] | | 0.30[***] | | -0.17[*] | | -0.05 | | 0.13[*] | | 0.09 | |
| Adaptability | 4.86 | 1.32 | 0.09 | | 0.07 | | 0.39[***] | | **0.50**[***] | | 0.33[***] | | -0.02 | | 0.13 | | 0.09 | | -0.09 | |
| Mood | 5.86 | 1.17 | 0.16[*] | | 0.01 | | 0.08 | | 0.24[*] | | **0.39**[***] | | 0.04 | | 0.11 | | 0.17[**] | | -0.06 | |
| Attention | 4.24 | 1.52 | -0.03 | | 0.11 | | 0.12 | | 0.20[**] | | 0.27[***] | | **-0.10** | | 0.05 | | 0.09 | | -0.17[*] | |
| Persistence | 4.25 | 1.36 | 0.06 | | 0.05 | | -0.14[*] | | -0.12 | | -0.15[*] | | 0.37[***] | | **0.40**[***] | | 0.15[*] | | 0.01 | |
| Sensitivity | 4.95 | 1.41 | -0.06 | | -0.18[**] | | 0.07 | | 0.01 | | -0.08 | | -0.04 | | -0.19[**] | | **-0.41**[***] | | -0.01 | |
| Intensity | 5.31 | 1.30 | 0.21[**] | | -0.10 | | -0.02 | | -0.07 | | -0.25[***] | | -0.05 | | -0.03 | | 0.00 | | **0.36**[***] | |

\* < .05

\*\* < .01

\*\*\* < .001.

dimension was expectable because we designed the question only reflecting one's attentional focusing while the original distractibility subscale of the BSQ also measured emotional distractibility. A discordant correlation was found for sensitivity dimension ($r = -0.41$, $p < .001$), because higher sensitivity item score of the TBCS short measure would mean lower threshold of sensory stimuli measured by the BSQ. For certain dimensions, the correlations did not meet the criterion of a strong effect size ($r = 0.50$) [23]. It is anticipatory given that temperament generally relies on multiple items to reflect child's behaviours and reactivity in various contexts and single-item measures are unable to fully capture such a complex construct.

## Data source and participants

The data came from the TBCS, a nationally representative longitudinal study that aims to depict the health profile of Taiwanese children and examine early origins of diseases during the developmental course until early adulthood. The study cohort initially comprised 24,200 live births drawn from the 2005 National Birth Report Database (sampling rate 11.7%) by two-stage stratified random sampling and the children were followed up with six waves of surveys before they entered secondary school [24]. All the surveys were conducted via face-to-face interviews with the child's mother or primary caregiver with their written informed consent.

The present investigation was based on a sample of 18,994 children who completed the first survey at 6 months (2005/6~2006/7) and follow-ups at ages 5.5 years (2010/6~2011/7) and 8 years (2013/4~2014/8). We used a set of demographic characteristics and child health variables at baseline to conduct attrition analysis. The results showed that this study sample was not statistically different from the initial cohort by sex, perinatal conditions (i.e. preterm birth, small-for-gestational age, congenital problems) and parent-reported health status. However, mothers of children lost to follow-up tended to be younger at child birth and with lower educational levels.

This study was approved by the research ethics committee of the National Health Research Institutes, Taiwan (No. EC1020102-F).

## Study variables

**Temperament traits.** Temperament traits at age 5.5 years were measured using the previously validated nine-item measure. By reviewing and comparing a set of broad factors that recent researchers have converged from various trait dimensions identified by major temperament models [3, 4, 11], we proposed two higher-order temperament factors- *surgency* and *regulation*. Surgency relates to positive emotions, approach behaviours and quick adaptation to new situations (i.e. mood, approach and adaptation). Because high activity levels are often associated with high levels of approach and tendency of positive emotions, activity dimension was also included under the surgency factor. Regulation encompasses the ability to pay attention, persist as well as modulate emotional and behavioural reactivity (i.e. attention, persistence, and sensitivity). While intensity of reactions is more appropriately put under a separate emotionality factor and rhythmicity has not been consistently listed under certain factors, we did not include these two factors in defining and testing higher-order temperament traits. Confirmatory factor analysis using Mplus 8.0 [25] demonstrated a modestly good model fit (CFI = 0.94; RMSEA = 0.08, 95% CI 0.074–0.080), providing support for its conceptually valid structure. Two scores were derived by summing the factor-specific item scores, with higher scores representing greater levels of temperament traits. The correlation coefficients between the items and temperament trait scores of the TBCS short measure are presented in S2 Table. We categorized temperament traits into three levels: low ($\leq 20^{th}$ percentile), medium ($> 20^{th}$ to $< 80^{th}$ percentile), and high ($\geq 80^{th}$ percentile).

**Physical health.** Children's physical health was reflected by caregiver-rated general health status and injury incidence using 8-year-old survey data, both operationalized as binary variables. The caregiver was asked to rate on a 5-point Likert scale as to the current health status of the child compared to others of the same age. This single item has been shown to be valid in reflecting the actual health of a child [26] and was further categorized into poor health (fair/poor/very poor) versus non-poor health (very good/good). Injury incidence referred to whether the child had ever sought medical attendance for a fall or other injury events over the past year.

**Covariates.** The covariates were mainly chosen by referring to the ecosystem model of child health: human biology (sex, adverse birth outcome, congenital conditions), personal behaviour (health behaviours such as washing hands, playing outdoor), socioeconomic and demographic environment (maternal age, maternal education) [27]. As injury represents a significant child health issue for which specific aetiological models and preventive measures have usually been identified and in place [28, 29], we included a slightly different set of adjusted covariates such as injury history and the frequency of engaging in moderate-vigorous physical activity. Adverse birth outcome was defined by child's preterm and small-for-gestational (SGA) status as a binary variable (yes = preterm, SGA or both; no = none of them). The child's health behaviours at age 8 were measured using four dichotomous variables (yes = 1, no = 0): always washing hands before meals and after using the toilet, eating fruits and vegetables every day, always/often playing outdoors, and watching TV less than two hours per day on weekdays. Health behaviour scores were calculated as the sum of the four item scores, with higher scores indicating healthier behavioural practices. Maternal education (junior high school or below, senior high school, and college and above) was used as a general socioeconomic variable indicative of family financial resources and caregiver health literacy.

**Data analysis.** We calculated descriptive statistics of children's temperament at age 5.5 stratified by sex and adverse birth outcomes. In order to demonstrate and test the linear trend of the distribution of physical health outcomes across different levels of temperament traits at age 5.5, cross-tabulations with the Cochran-Armitage chi-square test were performed. We further applied multivariate logistic regression analysis. Two dichotomous health variables were separately regressed on dummy variables for surgency and regulation as well as selected covariates. The odds ratios (ORs) of the predicted variables were estimated at a significance level of .05. All analyses were performed by using SAS software (version 9.4).

## Results

### Descriptive and bivariate analyses of children's temperament

As shown in Table 2, 52.4% of the study sample were boys, 8.3% were born preterm (gestational age < 37 weeks) and 9.4% were SGA births (gestation-adjusted birthweight below the 10th percentile). The majority of the children's mothers had attained a secondary (40.1%) or college or higher (45.3%) levels of education and the mean age was 36.9 years.

The general temperament profile of Taiwanese children at 5.5 years of age reflected by the nine-item temperament measure is presented in Table 3. The item means across all temperament dimensions were higher than the midpoint level (4) of the scales, with relatively lower scores for the attention, approach and response intensity dimensions. The mean score was 21.6 (SD = 3.6; range: 5–28) for the surgency trait and 15.0 (SD = 3.0; range 4–21) for the regulation trait. Significant sex differences were found, with boys demonstrating higher scores for activity level, approach and the surgency trait but lower scores for adaptability, attention, persistence, sensitivity and the regulation trait. Children born SGA scored significantly lower in rhythmicity, attention, persistence, sensitivity and regulation trait and higher in intensity. The

**Table 2. Descriptive statistics of the study variables.**

| Study variables | n | % |
|---|---:|---:|
| **Sex** | | |
| Boy | 9949 | 52.4 |
| Girl | 9045 | 47.6 |
| **Preterm birth** | | |
| Yes | 1581 | 8.3 |
| No | 17413 | 91.7 |
| **Small-for-gestational age (SGA) birth**[a] | | |
| Yes | 1785 | 9.4 |
| No | 17209 | 90.6 |
| **Maternal age** (*Mean*, *SD*) | 36.9 | 4.8 |
| **Maternal education** | | |
| Junior high school and below | 2715 | 14.3 |
| Senior high school | 7612 | 40.1 |
| College and above | 8637 | 45.4 |
| Missing | 30 | 0.2 |
| **Physical health outcomes at 8 years** | | |
| Parent-reported poor health[b] | 3866 | 20.4 |
| Medically attended injury | 4487 | 23.6 |
| **Health behaviours at 8 years** | | |
| Washes hands before meals and after using toilet | 7732 | 40.7 |
| Eats fruits and vegetables every day | 10380 | 54.7 |
| Always/often plays outdoors | 13598 | 71.6 |
| Watches TV less than two hours per weekday | 16122 | 84.9 |
| **Health behaviours score at 8 years** (Mean, SD) | 2.5 | 1.0 |
| Total | 18,944 | 100.0 |

[a] Defined as gestation-adjusted birthweight below the 10th percentile

[b] Defined as a parent rating of the child's health as fair, poor or very poor.

results were in line with previous studies showing consistent sex differences in the activity levels, approach, inhibitory control and sensitivity of preschool children [30, 31] and variations in toddlers' attention span and sensitivity by birth outcomes [32].

## Early temperament traits and physical health

In Table 4, the cross-tabulations show significant relationships of the temperament trait level at 5.5 years old with the parent-rated general health status at 8 years old but not with injury occurrence. More specifically, the percentage of children rated as having poor health significantly decreased as the surgency trait level increased by the Cochran-Armitage trend test, and a similar gradient was observed for the regulation trait.

Table 5 displays the results of the logistic regression models for predicting child's poor health and medically attended injury occurrence at age 8. Compared to children with low surgency trait levels, those with medium (OR = 0.84, 95% CI [0.77–0.93]) or high (OR = 0.71, 95% CI [0.63–0.80]) surgency trait levels were less likely to have poor health. Likewise, children with higher regulation trait levels (medium: OR = 0.76, 95% CI [0.69–0.83] and high: OR = 0.66, 95% CI [0.58–0.74]) had a lower probability of poor parent-rated health. While early surgency trait was not a significant predictor of medically attended injury occurrence,

**Table 3. Means and standard deviations of the temperament dimension and trait scores by child's basic characteristics (N = 18,994).**

| | Temperament dimension | | | | | | | | | Temperament traits | |
| | Activity | Rhythmicity | Approach | Adaptability | Mood | Attention | Persistence | Sensitivity | Intensity | Surgency | Regulation |
|---|---|---|---|---|---|---|---|---|---|---|---|
| **Total** | 5.6 (1.1) | 5.5 (1.2) | 4.9 (1.5) | 5.2 (1.3) | 5.8 (1.1) | 4.6 (1.4) | 5.1 (1.4) | 5.4 (1.2) | 4.9 (1.4) | 21.6 (3.6) | 15.0 (3.0) |
| **Sex** | | | | | | | | | | | |
| Boy | 5.8 (1.0)[a] | 5.5 (1.2) | 4.9 (1.4)[a] | 5.1 (1.2)[a] | 5.8 (1.1) | 4.4 (1.4) | 5.0 (1.4) | 5.2 (1.2) | 4.9 (1.4) | 21.8 (3.5) [a] | 14.7 (3.0) |
| Girl | 5.4 (1.1) | 5.5 (1.2) | 4.8 (1.5) | 5.2 (1.3) | 5.8 (1.1) | 4.7 (1.4) [a] | 5.1 (1.3) [a] | 5.5 (1.2) [a] | 4.9 (1.4) | 21.3 (3.6) | 15.3 (2.9) [a] |
| **Preterm birth** | | | | | | | | | | | |
| Yes | 5.6 (1.1) | 5.5 (1.2) | 4.9 (1.5) | 5.2 (1.3) | 5.8 (1.0) | 4.5 (1.4) | 5.0 (1.4) | 5.3 (1.2) | 5.0 (1.4) | 21.5 (3.7) | 14.9 (3.1) |
| No | 5.6 (1.1) | 5.5 (1.2) | 4.9 (1.4) | 5.2 (1.6) | 5.8 (1.1) | 4.6 (1.4) | 5.1 (1.3) | 5.4 (1.2) | 4.9 (1.4) | 21.6 (3.6) | 15.0 (3.0) |
| **SGA birth** | | | | | | | | | | | |
| Yes | 5.7 (1.1) | 5.4 (1.2) | 4.8 (1.5) | 5.2 (1.3) | 5.8 (1.1) | 4.4 (1.5) | 5.0 (1.4) | 5.3 (1.3) | 5.0 (1.4) [b] | 21.5 (3.6) | 14.6 (3.2) |
| No | 5.6 (1.1) | 5.5 (1.1) [b] | 4.9 (1.4) | 5.2 (1.3) | 5.8 (1.0) | 4.6 (1.4) [b] | 5.1 (1.3) [b] | 5.4 (1.2) [b] | 4.9 (1.4) | 21.6 (3.6) | 15.0 (3.0) [b] |

[a] & [b] denote significantly higher item or subscale scores than the comparison group ($p < .05$)

children with high regulation trait level at younger age were significantly less likely to have an injury (OR = 0.90, 95% CI [0.81–0.99]).

## Discussion

### Major findings

The significance of early temperament traits that account for individual variations in developmental outcomes and susceptibility to or protection against psychopathological problems has been well recognized. However, less evidence is available for the influences of temperament on children's physical health. Our study was to fill this gap by employing a longitudinal design with a large population-based sample.

We focused on two higher-order temperament traits—surgency and regulation that were theoretically sound and statistically verified. Both surgency and regulation traits appeared to be protective for children's general health. A high surgency trait level, by our definition, referred to children being more physically vigorous, being open to novelty, easily adapting to changes and in general showing positive affect. This construct is similar to the positive emotionality trait described by Hampson and Vollrath in their review, in which they applied

**Table 4. Cross-tabulations of early temperament traits and physical health at age 8 with the Cochran-Armitage trend test (N = 18,994).**

| Temperament at age 5.5 | Poor health | | | | | | Medically attended injury | | | | |
| | Total | | Yes | | No | | P value | Yes | | No | | P value |
| | N | % | N | % | n | % | | n | % | n | % | |
|---|---|---|---|---|---|---|---|---|---|---|---|---|
| **Surgency trait** | | | | | | | < .001 | | | | | 0.11 |
| Low | 3819 | 20.1 | 1012 | 26.5 | 2807 | 73.5 | | 868 | 22.7 | 2951 | 77.3 | |
| Medium | 11021 | 58.1 | 2204 | 20.0 | 8817 | 80.0 | | 2613 | 23.7 | 8408 | 76.3 | |
| High | 4141 | 21.8 | 644 | 15.6 | 3497 | 84.4 | | 1004 | 24.2 | 3137 | 75.8 | |
| **Regulation trait** | | | | | | | < .001 | | | | | 0.07 |
| Low | 3799 | 20.0 | 1085 | 28.6 | 2714 | 71.4 | | 922 | 24.3 | 2877 | 75.7 | |
| Medium | 11108 | 58.5 | 2164 | 19.5 | 8944 | 80.5 | | 2643 | 23.8 | 8465 | 76.2 | |
| High | 4080 | 21.5 | 611 | 15.0 | 3469 | 85.0 | | 922 | 22.6 | 3158 | 77.4 | |

Categories of temperament traits: low (≤20th percentile), medium (>20th to <80th percentile), and high (≥80th percentile)

**Table 5. Odds ratios from the logistic regression analysis of early temperament traits predicting physical health at age 8 (N = 18,994).**

|  | Poor health | Medically attended injuries |
|---|---|---|
|  | Odds Ratio [95% CI] | Odds Ratio [95% CI] |
| **Surgency trait at age 5.5** (ref = Low) |  |  |
| Medium | 0.84*** [0.77–0.93] | 1.05 [0.96–1.15] |
| High | 0.71*** [0.63–0.80] | 1.10 [0.99–1.22] |
| **Regulation trait at age 5.5** (ref = Low) |  |  |
| Medium | 0.76*** [0.69–0.83] | 0.97 [0.89–1.06] |
| High | 0.66*** [0.58–0.74] | 0.90* [0.81–0.99] |
| **Sex** (ref = Girl) |  |  |
| Boy | 1.18*** [1.09–1.27] | 1.17*** [1.09–1.25] |
| **Adverse birth outcomes** | 1.17** [1.06–1.28] | 1.08 [0.99–1.18] |
| **Congenital conditions** | 1.34*** [1.14–1.57] | 1.18* [1.02–1.37] |
| **Maternal age** (ref = < 30 years) |  |  |
| 30 ~40 years | 0.90 [0.77–1.04] | 1.03 [0.89–1.20] |
| > = 40 years | 0.93 [0.79–1.10] | 1.05 [0.90–1.23] |
| **Maternal education** (ref: < = Junior high school) |  |  |
| High school | 0.85** [0.76–0.95] | 1.15** [1.04–1.29] |
| > = College | 0.86** [0.76–0.96] | 1.30*** [1.16–1.44] |
| **Health behaviours scores at 8 years** | 0.82*** [0.79–0.85] | — |
| **Poor health at age 5.5 years** | 4.48*** [4.15–4.84] | — |
| **Injury occurrence at age 5.5 years** | — | 1.29*** [1.15–1.45] |
| **Frequency of physical activity** | — | 1.09*** [1.07–1.11] |
| R squared values | 0.106 | 0.016 |
| AUC values | 0.727 | 0.570 |

\* < .05

\*\* < .01

\*\*\* < .001.; AUC: area under the curve

reinforcement sensitivity theory to propose a biological mechanism linking positive emotionality and children's well-being [19]. Specifically, positive emotionality is associated with sensitivity to rewards, which reinforces the motivation to seek opportunities for appetitive stimuli and stress-relieving experiences [33], thereby mitigating physiological arousal and perceived stress. This may partly account for the protective effect of children's surgency traits on physical health. Moreover, evidence has indicated individual differences in cortisol reactivity in relation to temperamentally based regulation [1]. For example, generally higher cortisol levels or greater cortisol reactivity in frustrating situations were found in pre-schoolers who scored lower in the regulatory domains of temperament traits [34, 35]. Less than optimal cortisol reactivity may in turn lead to detrimental long-term health outcomes. However, constrained by using observational data, we were unable to ascertain the underlying biological mechanisms as the aforementioned evidence has suggested. In regard to medically attended injury, only the regulation trait showed a significant effect. This result was in accordance with prior studies in relation to children's control tendency but not surgency traits [16, 17]. The association between activity level and injury was well evidenced, yet our definition of surgency trait included other temperament dimensions that may have relatively small effects on unintentional injury. In addition, a measurement issue should be noted. As opposed to a rated measure of injury severity as used in another study [16], we only considered injuries that were

assessed and treated in medical settings of any kind and whether or not an injury occurred. Nevertheless, a consensus seems to be reached that a child's temperament characterized by a better attention span and being more sensitive to external stimuli can reduce the risk for injury.

## Strengths and limitations

One of the strengths of the current study is that we applied a longitudinal design, through which the contribution of early temperament to later physical health could be verified given the possible reciprocal relations between personality and health [19]. Second, for this purpose, we validated a temperament measure based on constructs derived from the NYLS with only nine items, and the measure showed acceptable correlations with the original scales and established known-group validity (i.e., significant differences in item scores by sex and birth outcome status). Although evidence of the reliability of the instrument is lacking, the use of a 7-point Likert scale with several vignettes provided in the statements could improve the reliability. This short instrument is appropriate for use in surveys with consideration of participant burden. There is also a potential for its practical applications in childcare and health, as suggested that temperament measures be incorporated into early childhood assessments to provide parents and childcare providers with useful information for fostering their caregiving experiences [36]. Finally, our temperament measure was built upon Thomas and Chess's behavioural style model but can be operationalized into two higher-order traits that roughly correspond to the constructs of other temperament models, such as surgency and effortful control, identified by Rothbart and colleagues [1]. This work has somewhat echoed the thinking of temperament researchers that assessments should be created by encompassing the constructs from different traditions to reflect both the common set and unique aspects of temperament traits [2, 37].

The results should be interpreted in view of some limitations. The first concerns using a caregiver-report measure of temperament. Mothers or caregivers may be biased in their perceptions or expectations of child's behaviours and reactivity [3]. Second, while we relied solely on caregiver report of both temperament traits and health outcomes of the study cohort, their relations could have been affected by shared measurement errors. Third, we failed to adjust for certain factors which may have confounded the findings. For example, the presence of health conditions or functioning problems can be a crucial component in caregiver rating of child's general health status. Injury risk behaviours and exposure to environmental hazards in any forms were also omitted from the model predicting child injury occurrence. Future research should consider using the short temperament measure with multiple informants or in conjunction with other assessment methods. Besides, objective measures of children's physical health outcomes such as by linking administrative health data could also be obtained to test the hypothesis.

## Conclusions

Our findings contribute new insights into the relations between early temperament and the physical health of school-age children. Specifically, higher levels of surgency and regulation traits in early life are protective of poor health status. Nonetheless, much remains undetermined about the linkages of these variables through a complex set of multifaceted processes. Of note is that temperament traits develop and change in a transactional manner, the effects of temperament-environment interactions along the developmental course should also be explicitly addressed [38]. Future empirical investigations in this area will have valuable implications

for creating health- and safety-promoting environments or interventions in response to child temperament characteristics.

## Supporting information

**S1 Table. The dimensions and corresponding questions of the TBCS short temperament measure.**
(DOCX)

**S2 Table. Correlation coefficients between the items and temperament trait scores of the TBCS short measure (N = 18,944).**
(DOCX)

## Acknowledgments

The authors wish to thank the parents and children for their time and long-term support participating in the Taiwan Birth Cohort Study.

## Author Contributions

**Conceptualization:** Jennifer Chun-Li Wu, Pei-Ling Wang, Tung-liang Chiang.

**Data curation:** Jennifer Chun-Li Wu.

**Formal analysis:** Jennifer Chun-Li Wu.

**Funding acquisition:** Tung-liang Chiang.

**Investigation:** Pei-Ling Wang.

**Methodology:** Jennifer Chun-Li Wu.

**Supervision:** Pei-Ling Wang.

**Validation:** Jennifer Chun-Li Wu, Tung-liang Chiang.

**Writing – original draft:** Jennifer Chun-Li Wu.

**Writing – review & editing:** Pei-Ling Wang, Tung-liang Chiang.

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
