## [Decision Letter · Decision Letter 0]

10 Oct 2022

PONE-D-21-36619Early temperament and physical health in school-age children: Applying a short temperament measure in a population-based cohortPLOS ONE

Dear Dr. Wang,

Thank you for submitting your manuscript to PLOS ONE. After careful consideration, we feel that it has merit but does not fully meet PLOS ONE’s publication criteria as it currently stands. Therefore, we invite you to submit a revised version of the manuscript that addresses the points raised during the review process.

We look forward to receiving your revised manuscript.

Kind regards,

Chong Chen

Academic Editor

PLOS ONE

Journal Requirements:

Reviewers' comments:

Reviewer's Responses to Questions

**Comments to the Author**

1. Is the manuscript technically sound, and do the data support the conclusions?

Reviewer #1: Partly

Reviewer #2: Yes

2. Has the statistical analysis been performed appropriately and rigorously? 

Reviewer #1: No

Reviewer #2: Yes

3. Have the authors made all data underlying the findings in their manuscript fully available?

Reviewer #1: No

Reviewer #2: Yes

4. Is the manuscript presented in an intelligible fashion and written in standard English?

Reviewer #1: Yes

Reviewer #2: Yes

5. Review Comments to the Author

Reviewer #1: The authors reported the relation between temperament at age 5.5 years measured by a short home-developed questionnaire and physical health at age 8 years.

The manuscript is of interest, however, several considerations emerged.

- The introduction is quite long, however, as there are already several scale for temperament measurement in young children, the need for a new scale is not obvious and should be more explicit.

- Regarding the new scale, how was selected the one item per dimension? All items and the corresponding question of the new scale should be provided.

- What was the order for questionnaire completion by parents? Was the new one always completed after the BSQ? What could be the consequences regarding the answers and subsequent results?

- How was calculated the new score?

- Who were the mother included in the new scale “validation study”? Do they have the same characteristics than the TBCS population? If no, what could be the implication for the interpretation of the result

- The mean score and SD should be provided for all items and subscales in table 1.

- The correlations between the new questionnaire items and the corresponding ones from the BSQ are significantly different from zero but still quite low (<0.7) considering that the authors want to measure the same phenotype. This should be more discussed.

- On the same line, discussion on discordant correlations should be added (e.g., sensitivity/persistence).

- Was the TBCS population included in the current analysis different from the one of the TBCS initial population? In other term, is there any inclusion bias in the studied population? If yes, how this was accounted for?

- How covariates for adjusted models were chosen? This need to be more detailed since some important confounders are missing.

- Why the threshold of 2500g was chosen for low birthweight? Were there premature children? Were they small for gestational age?

- The grouping in higher order temperament factor should be better explained. Why use only 7 out of the 9 items? What were the criteria to group them? This should be detailed and a table of between-items correlations should be provided.

- What are the correlations between the new score items and the temperament traits?

- Was the question on general health status at age 5.5 years the same than at age 8 years?

- Were specific interactions tested? Arguments for stratified analysis should be provided.

- Analyses reported in table 4 are chi-square tests, showing different distribution of temperament trait according to health and injuries. The tests do not allow conclusion on decreasing or increasing percentages except if the p-value reported is a tendency one. If yes, this should be detail in the method section.

- Tables 5 and 6, the number of included children in each analysis should be added.

- The discussion section should be revised according to previous comments. Term such as “proven” “demonstrated” should be nuanced.

- The current results do not provide “possible explanations for their theoretical rationales” as written page 21. This should be modified and/or better explained

Reviewer #2: The present study investigates the relationship between early temperament traits and physical health in a cohort of 18,994 school-age children. Higher levels of early surgency and regulation traits, emerged as predictors of lower caregiver-rated poor health No trait was associated with higher injury risk in the global sample; higher levels of regulation reduced injury occurrence risk in boys only.

The study is timely and reports interesting findings. The paper is well-written, and the analyses are appropriate for the study design.

Overall, the study represents a worthy contribution to the research field.

However, some minor revisions could further improve the overall interest for the reader.

Overall: While the paper is well written and easy to read, it should be spell-checked as some minor English errors/spelling mistakes can ben found throughout the manuscript.

Methods: The characteristics of the sample reported in Table 2 should be moved form the Methods section to the beginning of the Results section.

Methods and Results: Three models for each outcome appears to be an excessive number of analyses, particularly without a correction for family-wise errors. I suggest to either describe the separate regression models specific for each gender as supplementary analyses, or to introduce gender as a proxy variable in the total sample regression models and assess if it represents a significant predictor for that outcome.

Results: Beside Odds-ratios, raw R2 values of regression analyses should be reported in the manuscript.

Discussion: While future research perspectives are briefly mentioned in the final sentence of the manuscript, a more extended, dedicated paragraph at the end of the Discussion section could increase the interest for the reader.

6. PLOS authors have the option to publish the peer review history of their article (what does this mean?). If published, this will include your full peer review and any attached files.

Reviewer #1: No

Reviewer #2: No

---

## [Author Response · Author response to Decision Letter 0]

14 Jan 2023

The comments of the reviewers are truly helpful, and we took them very seriously in revising the manuscript. Overall, the revision focuses on: (1) clarifying in the Introduction the reason and contribution of validating a new temperament measure; (2) providing more details about the validation procedure of the short measure and how the structure of temperament traits was determined in the Methods section; (3) modifying the logistic regression models and the results accordingly; and (4) strengthening the discussion of our study limitations and implications for future research. Point-by-point responses are provided below, and the changes in the manuscript are highlighted in yellow. 

Reviewer #1: The authors reported the relation between temperament at age 5.5 years measured by a short home-developed questionnaire and physical health at age 8 years. The manuscript is of interest, however, several considerations emerged.

1. The introduction is quite long, however, as there are already several scale for temperament measurement in young children, the need for a new scale is not obvious and should be more explicit.

Response: Thank you for this important comment. The objective of this study is to examine whether early temperament traits predict physical health of school-age children. The introduction thus has mainly drawn on existing literature to address this research gap. Because our analysis was based on data from a large-scale longitudinal study - Taiwan Birth Cohort Study (TBCS), it became necessary for us to validate the short temperament measure designed to fit into the TBCS questionnaire. In spite of that, we also considered it valuable to develop the TBCS short measure given that temperament instruments tend to be lengthy and time-consuming to administer, yet short forms remain limited. A short scale is not only more applicable for use in large-scale observational studies but also useful for practical purposes such as suggested that temperament measures should be included as part of early childhood assessments (citation [23]. We have reorganized the paragraphs in the Introduction section to strengthen this notion. (lines 90-99) 

2. Regarding the new scale, how was selected the one item per dimension? All items and the corresponding question of the new scale should be provided.

Response: Thank you. The one item per dimension of the TBCS short scale was not selected from the original Behavioural Style Questionnaire (BSQ). Instead, each item was initially written by the second author with reference to the BSQ which contains multiple items measuring the child’s behavioural style in interaction with the environment across situations. The second author is an academic expert in child development and early childhood education and has had extensive research experiences and several publications on child temperament. The nine items developed were further reviewed by the TBCS research team. The dimensions and corresponding questions have been provided in the supporting information S1 Table. 

3. What was the order for questionnaire completion by parents? Was the new one always completed after the BSQ? What could be the consequences regarding the answers and subsequent results?

Response: Thank you for bringing up this measurement issue. For the validation study, the questionnaire was structured into three parts: demographic data, the TBCS short measure, followed by the BSQ. Because the TBCS short measure was placed to precede the BSQ, we assume that overall impression of child’s behaviour style reported by the caregivers could be appropriately captured and was least likely to be affected by the BSQ which consisted of multiple specific questions. However, this may lead to weaker correlations between the item scores and BSQ subscale scores. We have modified the description to show in what order the scales were presented in the questionnaire. (lines 125-126)

4. How was calculated the new score?

Response: The TBCS short measure was rated on a 7-point scale, ranging from 1 to 7. We calculated item mean scores as presented in Table 1 and described in the text. 

5. Who were the mother included in the new scale “validation study”? Do they have the same characteristics than the TBCS population? If no, what could be the implication for the interpretation of the result.

Response: Thank you for pointing out this possible sample selection. For the validation study, we recruited a convenience sample of 228 children (mean age=59.8 months) from 10 preschools located in two administration areas- Taipei city and New Taipei city. The mothers of this sample were with a mean age of 35.4 years (SD=4.7), comparable to that of the TBCS cohort at 34.9 years (SD=4.8). More than half (54.9%) held a college or higher degree while 32.0% completed senior high school, showing higher educational attainment than their TBCS counterparts (45.4% college or higher, 40.1% senior high school). Although the validation study was based on a child sample whose mothers had higher levels of education, we would expect it not to affect our results of the validation. This is because both scales (i.e. short temperament scale and BSQ) mainly inquired about child’s behaviours that can be observed across different situations in everyday life and response biases potentially related to socioeconomic factors such as professional knowledge and social desirability were less likely to occur. 

6. The mean score and SD should be provided for all items and subscales in table 1.

Response: Thank you. The mean scores and SD of all items of the TBCS short measure and BSQ subscales have been provided in Table 1.

7. The correlations between the new questionnaire items and the corresponding ones from the BSQ are significantly different from zero but still quite low (<0.7) considering that the authors want to measure the same phenotype. This should be more discussed.

Response: Thank you for the comment. It is ideal to have a strong correlation (r from 0.5 to 1.0) between the full and short forms to support that a short form is valid (citation [26]). Indeed, certain items of the short temperament measure were significantly correlated with the corresponding subscales (except attention dimension) but only in the range of medium effect sizes. It is anticipatory because single-item measures are unable to fully capture complex constructs, and child temperament is one example that multiple items are usually used to rate child’s behaviours across different situations in everyday life. We have discussed this issue in the revision (lines 137-140). Despite this concern, the TBCS short scale is acceptably valid by also establishing expert validity and known-group validity (e.g. with child gender) in this study. 

8. On the same line, discussion on discordant correlations should be added (e.g., sensitivity/persistence).

Response: Thank you. Although all correlations are illustrated in Table 1, we only reported those on the diagonal- that is, between the item score of the TBCS short measure and the corresponding BSQ subscales to provide evidence of validity. A discordant correlation was found for sensitivity dimension (r=-0.41, p<.001), because higher sensitivity item score of TBCS short scale would mean lower threshold of sensory stimuli measured by the BSQ. This has been described in the Methods. (lines 134-136) 

9. Was the TBCS population included in the current analysis different from the one of the TBCS initial population? In other term, is there any inclusion bias in the studied population? If yes, how this was accounted for?

Response: This current study only included children with data available for all three waves of surveys (6 months, 5.5 years and 8 years). We used a set of demographic characteristics and child health variables at baseline (6 months) to conduct attrition analysis. The results showed that this study sample was not statistically different from the initial cohort by sex, perinatal conditions (i.e. preterm birth, small-for-gestational age, congenital problems) and parent-reported health status. However, mothers of children lost to follow-up tended to be younger at child birth and with lower educational levels. We have reported sample attrition in the Methods section (lines 153-158). Since sample attrition in relation to selected child health variables was insignificant, the possibility of leading to biased estimates was not further discussed. 

10. How covariates for adjusted models were chosen? This need to be more detailed since some important confounders are missing.

Response: Thank you. The covariates adjusted in the regression models were chosen by referring to the ecosystem model of child health: human biology (sex, adverse birth outcome, congenital conditions), personal behaviour (health behaviours such as washing hands, playing outdoor) and socioeconomic environment (maternal education) (citation [30]). By reviewing additional literature on child injury risk (citation [31,32]), we also included the frequency of engaging in moderate-vigorous physical activity as a confounder in the regression models for injury. This has been described with more details in the Methods. (lines 195-201) 

11. Why the threshold of 2500g was chosen for low birthweight? Were there premature children? Were they small for gestational age? 

Response: Thank you for raising these questions. We initially included low birthweight (LBW) status in this study because infants born at the lower extreme of the birthweight distribution are at a higher risk for newborn and later morbidity. The criterion of 2500g has long been applied as a single LBW measure which allows for consistent reporting of vital statistics and between-country comparison. However, such an absolute cutoff may mask the actual maturity (gestational age) and fetal growth of the infants at birth. Therefore, in the revision, we identified children born preterm (gestational age < 37 weeks) or small-for-gestational age (SGA; gestation-adjusted birthweight below the 10th percentile) and reported the prevalence rates in Table 2. We further created a binary variable- adverse birth outcome (yes= preterm, SGA or both; no=none of them) to be included in the logistic regression models as a covariate. We have made changes accordingly in all appropriate instances. 

12. The grouping in higher order temperament factor should be better explained. Why use only 7 out of the 9 items? What were the criteria to group them? This should be detailed and a table of between-items correlations should be provided.

Response: Thank you. There is generally a consensus defining temperament as early-appearing individual differences in traits that are relatively consistent across situations, moderately stable and under genetic influences. However, the structures of temperament postulated by several theoretical models (e.g. Thomas and Chess, Rothbart, Buss and Plomin) have varied with respect to the dimensions and descriptions of core traits. More recent researchers have attempted to converge the various dimensional traits into a set of broad factors, such as: emotionality, extraversion, activity, and persistence (citation [4]); emotions, activity and attention/regulatory behaviours (citation [3]); extraversion/surgency, negative emotionality, sociability, activity level, regularity ability (citation [11]). By comparing the dimensions that fall within these broad factors and in consideration of the dimensions measured with the TBCS short scale, we then proposed two higher-order temperament factors named surgency and regulation, respectively. Surgency relates to positive emotions, approach behaviours and quick adaptation to new situations (i.e. mood, approach and adaptation). Because high activity levels are often associated with high levels of approach and tendency of positive emotions, activity dimension was also included under the surgency factor. Regulation encompasses mainly the ability to pay attention, persist as well as modulate emotional and behavioural reactivity (i.e. attention, persistence, and sensitivity). While intensity of reactions is more appropriately put under a separate emotionality factor and rhythmicity has not been consistently listed under certain factors, we did not include these two factors in the defining and testing higher-order temperament traits. The criteria of how two higher-order temperament traits were composed have been described in more detail (lines 167-178) and between-item correlations have been presented in S2 Table. 

13. What are the correlations between the new score items and the temperament traits?

Response: Thank you. Also shown in S2 Table., the surgency trait has strong positive correlations with the items composing the trait score- activity (r=0.62), approach (r=0.83), adaptability (r=0.81) and mood (r=0.65) at a significance level (p<.001). Likewise, strong positive correlations are found between regulation trait and attention (r=0.80), persistence (r=0.79) and sensitivity (r=0.68). 

14. Was the question on general health status at age 5.5 years the same than at age 8 years?

Response: Thank you. Yes. The same question- “In general, would you say your child’s health is very good, good, fair, poor or very poor?” was used in both surveys at age 5.5 years and 8 years to allow for longitudinal follow-up. 

15. Were specific interactions tested? Arguments for stratified analysis should be provided.

Response: Thank you. With the main research question concerning the relationship between early temperament traits and later physical health, we did not intend to examine any moderating effects. It might have been confusing since in the previous version of the manuscript, we presented the results of logistic regression analysis stratified by child’s gender. This was also criticized by the other reviewer for possibly making Type I error with testing multiple hypotheses (family-wise error). In the current version, we have revised Table 5 which reflects newly calculated logistic regression models for two child health outcomes, based on the total sample without stratified analysis and with no interaction effects tested.

16. Analyses reported in table 4 are chi-square tests, showing different distribution of temperament trait according to health and injuries. The tests do not allow conclusion on decreasing or increasing percentages except if the p-value reported is a tendency one. If yes, this should be detail in the method section. 

Response: Thank you. In order to test the linear trend of the distribution of child health and injuries across different levels of temperament traits, the Cochran-Armitage chi-square test was used instead. This has been described in the Method section (lines 242-244). 

17. Tables 5 and 6, the number of included children in each analysis should be added.

Response: Thank you. All children were included in both regression models and we have added the number in Table 5. 

18. The discussion section should be revised according to previous comments. Term such as “proven” “demonstrated” should be nuanced.

Response: Thank you. We have carefully checked and changed the wording in the Discussions section to make the notions more appropriately echo the results and limitations. (lines 297-299)

19. The current results do not provide “possible explanations for their theoretical rationales” as written page 21. This should be modified and/or better explained.

Response: We agree with you on this comment. The sentence has now been modified as “However, constrained by using observational data, we were unable to ascertain the underlying biological mechanisms as the aforementioned evidence has suggested” to provide better clarity of our argument.

Reviewer #2: The present study investigates the relationship between early temperament traits and physical health in a cohort of 18,994 school-age children. Higher levels of early surgency and regulation traits, emerged as predictors of lower caregiver-rated poor health. No trait was associated with higher injury risk in the global sample; higher levels of regulation reduced injury occurrence risk in boys only. The study is timely and reports interesting findings. The paper is well-written, and the analyses are appropriate for the study design. Overall, the study represents a worthy contribution to the research field. However, some minor revisions could further improve the overall interest for the reader.

1. Overall: While the paper is well written and easy to read, it should be spell-checked as some minor English errors/spelling mistakes can be found throughout the manuscript.

Response: Thank you. The spelling has been carefully checked and corrected throughout the manuscript.

2. Methods: The characteristics of the sample reported in Table 2 should be moved from the Methods section to the beginning of the Results section. 

Response: Thank you. We have moved the description of the participants’ characteristics to the beginning of the Results section. (lines 222-225)

3. Methods and Results: Three models for each outcome appears to be an excessive number of analyses, particularly without a correction for family-wise errors. I suggest to either describe the separate regression models specific for each gender as supplementary analyses, or to introduce gender as a proxy variable in the total sample regression models and assess if it represents a significant predictor for that outcome.

Response: Thank you for the suggestion. With the main research question concerning the relationship between early temperament traits and later physical health, we did not intend to examine any moderating effects. It might have been confusing since in the previous version of the manuscript, we presented the results of logistic regression analysis stratified by child’s gender. We also agree with you that such a stratified analysis would create Type I error with testing multiple hypotheses. In the current version, we have revised Table 5 which reflects newly calculated logistic regression models for two child health outcomes, based on the total sample without stratified analysis and taking child’s gender as a predictor variable.

4. Results: Beside Odds-ratios, raw R2 values of regression analyses should be reported in the manuscript.

Response: Thank you. We have reported R-squared and AUC (area under the curve) values of the logistic regression models in Table 5. 

5. Discussion: While future research perspectives are briefly mentioned in the final sentence of the manuscript, a more extended, dedicated paragraph at the end of the Discussion section could increase the interest for the reader.

Response: Thank you for the suggestions. We have added another paragraph in the Discussion section to address more delicately our study limitations and implications for future research. (lines 312-320)

---

## [Decision Letter · Decision Letter 1]

16 Feb 2023

PONE-D-21-36619R1Early temperament and physical health in school-age children: Applying a short temperament measure in a population-based cohortPLOS ONE

Dear Dr. Wang

Thank you for submitting your manuscript to PLOS ONE. After careful consideration, we feel that it has merit but does not fully meet PLOS ONE’s publication criteria as it currently stands. Therefore, we invite you to submit a revised version of the manuscript that addresses the points raised during the review process.

ACADEMIC EDITOR: As much as I would like to commend the authors for the corrections made to improve the manuscript as it stands, I agree with reviewer 1; there are some technical issues, especially regarding the introduction and methods, that need to be fixed before we can consider the manuscript suitable for publication. This is a major issue, mainly as it affects these manuscript portions. Therefore, I would like to give the author another opportunity to address this before making a final decision on the submission.

We look forward to receiving your revised manuscript.

Kind regards,

Anthony A. Olashore, MBCHB, FWACP

Academic Editor

PLOS ONE

Reviewers' comments:

Reviewer's Responses to Questions

**Comments to the Author**

1. If the authors have adequately addressed your comments raised in a previous round of review and you feel that this manuscript is now acceptable for publication, you may indicate that here to bypass the “Comments to the Author” section, enter your conflict of interest statement in the “Confidential to Editor” section, and submit your "Accept" recommendation.

Reviewer #1: (No Response)

Reviewer #2: All comments have been addressed

Reviewer #3: (No Response)

2. Is the manuscript technically sound, and do the data support the conclusions?

Reviewer #1: Yes

Reviewer #2: Yes

Reviewer #3: Yes

3. Has the statistical analysis been performed appropriately and rigorously? 

Reviewer #1: Yes

Reviewer #2: Yes

Reviewer #3: Yes

4. Have the authors made all data underlying the findings in their manuscript fully available?

Reviewer #1: No

Reviewer #2: Yes

Reviewer #3: Yes

5. Is the manuscript presented in an intelligible fashion and written in standard English?

Reviewer #1: No

Reviewer #2: Yes

Reviewer #3: Yes

6. Review Comments to the Author

Reviewer #1: The authors have taken most of my comments into account. However, some of their responses have raised new comments that must be addressed before the manuscript is suitable for publication.

The modification made in the introduction need to be revised. Indeed, it is not because you used large scale longitudinal data that you need to validate the questionnaire that was implemented. You chose in the design of TBCS to implement a new questionnaire. You need to validate it before analyzing it.

I understand the usefulness of using a short scale in large cohort however, the explanation provided does not really show any plus in using this scale compared to other already existing short ones.

The added sentence on the effect size in the method section lines 138-140“It is anticipatory given that single-item measures are unable to fully capture complex constructs and temperament is one example that generally relies on multiple items to reflect child’s behaviours and reactivity in various contexts” is confusing since temperament is the main topic of this validation study and you write lines 130-131 “ … providing evidence for the applicability of the nine-item temperament measure”. This should be rephrased.

I understand the interest in using higher order temperament factors. However, the entire introduction section focused toward this new 9-item scale and nothing prepares for the higher order factors that are finally the main results presented. Thus, as written, it seems like because nothing came out at the first analysis using the nine items you finally group them to hopefully find associations with health outcomes.

Why not the same adjustment factors in both models? Why the maternal age was not included? The observed relations can be affected by shared measurement errors but not only. There are some confounders that were not considered, this should be acknowledged and discussed in the limitation section.

Minor

Still many typos, e.g., temperature, table 5 age 5 and 5.5…

Reviewer #2: (No Response)

Reviewer #3: (No Response)

---

## [Author Response · Author response to Decision Letter 1]

24 Feb 2023

We would like to thank the reviewers for raising a number of critical points. All these have helped us to ensure that the main study objective holds while the reason of validating a short temperament measure is made clear. Point-by-point responses are provided below, and the changes in the revised manuscript are highlighted in blue. 

Reviewer #1: The authors have taken most of my comments into account. However, some of their responses have raised new comments that must be addressed before the manuscript is suitable for publication.

1. The modification made in the introduction need to be revised. Indeed, it is not because you used large scale longitudinal data that you need to validate the questionnaire that was implemented. You chose in the design of TBCS to implement a new questionnaire. You need to validate it before analyzing it.

Response: Thank you. We have made it clear that the need and purpose of validating the short temperament measure of TBCS is to apply it for further investigation of our research question. (lines 90-93)

2. I understand the usefulness of using a short scale in large cohort however, the explanation provided does not really show any plus in using this scale compared to other already existing short ones.

Response: Thank you and we agreed with you on this comment. The sentences in the Introduction that we attempted to explain the usefulness of developing a new short scale of temperament have been removed. 

3. The added sentence on the effect size in the method section lines 138-140“It is anticipatory given that single-item measures are unable to fully capture complex constructs and temperament is one example that generally relies on multiple items to reflect child’s behaviours and reactivity in various contexts” is confusing since temperament is the main topic of this validation study and you write lines 130-131 “ … providing evidence for the applicability of the nine-item temperament measure”. This should be rephrased.

Response: Thank you. The paragraph has been rephrased as “It is anticipatory given that temperament generally relies on multiple items to reflect child’s behaviours and reactivity in various contexts and single-item measures are unable to fully capture such a complex construct.” to avoid confusion. (lines 141-143)

4. I understand the interest in using higher order temperament factors. However, the entire introduction section focused toward this new 9-item scale and nothing prepares for the higher order factors that are finally the main results presented. Thus, as written, it seems like because nothing came out at the first analysis using the nine items you finally group them to hopefully find associations with health outcomes.

Response: We chose to derive higher-order factors in our analysis. The decision was made by comparing the advantages of using higher-order versus lower-order temperament traits and considering the plausibility of how temperament is related to physical health outcomes of interest. Thank you for reminding us of missing this piece of information which we have added in the Introduction. (lines 94-102)

5. Why not the same adjustment factors in both models? Why the maternal age was not included? The observed relations can be affected by shared measurement errors but not only. There are some confounders that were not considered, this should be acknowledged and discussed in the limitation section.

Response: Thank you. The ecosystem model of child health provided us with an overall framework for establishing the regression models. Child injury is an important health issue which specific etiological models and preventive measures are usually built for. Thus, we considered it more appropriate to include a slightly different set of covariates in the model for injury occurrence. How the covariates were selected has been modified and described in more detail. (lines 194-199). The question about maternal age is also well noted, given that mother’s age may have an influence on their perception of child’s health, childrearing attitudes and practices. In the newly calculated models, we included maternal age as a covariate and the main results remained unchanged. (Table 5.) Some potential confounders omitted from the analysis have been acknowledged in the Discussion. (lines 318-322)

6. Still many typos, e.g., temperature, table 5 age 5 and 5.5…

Response: Thank you. We have checked the text and tables carefully throughout the manuscript to avoid typos. 

Reviewer #2: None

Reviewer #3:

1. You spelt out a disorder associated with externalizing behaviors – ADHD, what would be a disorder associated with internalizing behaviors “ Depression, Adjustment etc” 

Response: Thank you. According to the references cited (citation [3,4]), we added in the text “anxiety or depression” which are psychopathological problems associated with internalizing behaviours and most addressed in the temperament literature. (lines 65-66) 

2. It would be clearer/easier for a reader if you included years during which follow-ups were done.

Response: Thank you. We have specified the period of data collection for the baseline and each follow-up surveys. (lines 152-153)　 

3. This study “was” delete “has been” 

Response: Thank you. The grammatical error was corrected. (line 159)

4. Remove “ay” 

Response: “ay” is a typo for “at” and has been corrected. (line 252)

5. How did you take care of a response from a caregiver in lower social strata Vis one from higher socioeconomic strata, regarding possible differences in evaluating what constitutes a good or poor health? (line188-189)

Response: Thank you. Indeed, there is evidence that individuals from different socioeconomic groups (e.g. education or income levels) may rate their health differently due to differential reporting standards or the content of health evaluated. Similar report patterns may also exist when caregivers evaluate child’s health on a subjective rating scale. By adjusting for caregiver's level of education in the regression models, we assumed this problem could be somewhat mitigated. 

6. Take care of typos e.g. could delete “such as” or insert “measurements” to read as ** Measurements such as** (line 50)

Response: Thank you. We have checked the text and tables carefully throughout the manuscript to avoid typos.

---

## [Decision Letter · Decision Letter 2]

2 May 2023

Early temperament and physical health in school-age children: Applying a short temperament measure in a population-based cohort

PONE-D-21-36619R2

Dear Dr. Wang,

We’re pleased to inform you that your manuscript has been judged scientifically suitable for publication and will be formally accepted for publication once it meets all outstanding technical requirements.

Kind regards,

Anthony A. Olashore, MBCHB, PhD

Academic Editor

PLOS ONE

Additional Editor Comments (optional):

Reviewers' comments:

Reviewer's Responses to Questions

**Comments to the Author**

1. If the authors have adequately addressed your comments raised in a previous round of review and you feel that this manuscript is now acceptable for publication, you may indicate that here to bypass the “Comments to the Author” section, enter your conflict of interest statement in the “Confidential to Editor” section, and submit your "Accept" recommendation.

Reviewer #1: All comments have been addressed

2. Is the manuscript technically sound, and do the data support the conclusions?

Reviewer #1: Yes

3. Has the statistical analysis been performed appropriately and rigorously? 

Reviewer #1: Yes

4. Have the authors made all data underlying the findings in their manuscript fully available?

Reviewer #1: Yes

5. Is the manuscript presented in an intelligible fashion and written in standard English?

Reviewer #1: Yes

6. Review Comments to the Author

Reviewer #1: (No Response)

7. PLOS authors have the option to publish the peer review history of their article (what does this mean?). If published, this will include your full peer review and any attached files.

Reviewer #1: No

---

## [Editor Report · Acceptance letter]

12 May 2023

PONE-D-21-36619R2 

Early temperament and physical health in school-age children: Applying a short temperament measure in a population-based cohort 

Dear Dr. Wang:

I'm pleased to inform you that your manuscript has been deemed suitable for publication in PLOS ONE. Congratulations! Your manuscript is now with our production department. 

Kind regards, 

on behalf of

Dr. Anthony A. Olashore 

Academic Editor

PLOS ONE